# Biophysical Characterization of Novel DNA Aptamers against K103N/Y181C Double Mutant HIV-1 Reverse Transcriptase

**DOI:** 10.3390/molecules27010285

**Published:** 2022-01-03

**Authors:** Siriluk Ratanabunyong, Supaphorn Seetaha, Supa Hannongbua, Saeko Yanaka, Maho Yagi-Utsumi, Koichi Kato, Atchara Paemanee, Kiattawee Choowongkomon

**Affiliations:** 1Department of Biochemistry, Faculty of Science, Kasetsart University, Bangkok 10900, Thailand; ae.med@hotmail.com (S.R.); Supaporn.se@ku.th (S.S.); 2National Omics Center, National Science and Technology Development Agency (NSTDA), Pathum Thani 12120, Thailand; atchara.pae@nstda.or.th; 3Department of Chemistry, Faculty of Science, Kasetsart University, Bangkok 10900, Thailand; fscisph@ku.ac.th; 4Center for Advanced Studies in Nanotechnology for Chemical, Food and Agricultural Industries, KU Institute for Advanced Studies, Kasetsart University, Bangkok 10900, Thailand; 5Exploratory Research Center on Life and Living Systems (ExCELLS) and Institute for Molecular Science (IMS), National Institutes of Natural Sciences, Okazaki 444-8787, Japan; saeko-yanaka@ims.ac.jp (S.Y.); mahoyagi@ims.ac.jp (M.Y.-U.); kkatonmr@ims.ac.jp (K.K.)

**Keywords:** HIV-1 RT, DNA aptamer, K103N/Y181C double mutant, gold nanoparticles, SPR, NMR, cytotoxicity and pseudo-HIV particles

## Abstract

The human immunodeficiency virus type-1 Reverse Transcriptase (HIV-1 RT) plays a pivotal role in essential viral replication and is the main target for antiviral therapy. The anti-HIV-1 RT drugs address resistance-associated mutations. This research focused on isolating the potential specific DNA aptamers against K103N/Y181C double mutant HIV-1 RT. Five DNA aptamers showed low IC50 values against both the KY-mutant HIV-1 RT and wildtype (WT) HIV-1 RT. The kinetic binding affinity forms surface plasmon resonance of both KY-mutant and WT HIV-1 RTs in the range of 0.06–2 μM and 0.15–2 μM, respectively. Among these aptamers, the KY44 aptamer was chosen to study the interaction of HIV-1 RTs-DNA aptamer complex by NMR experiments. The NMR results indicate that the aptamer could interact with both WT and KY-mutant HIV-1 RT at the NNRTI drug binding pocket by inducing a chemical shift at methionine residues. Furthermore, KY44 could inhibit pseudo-HIV particle infection in HEK293 cells with nearly 80% inhibition and showed low cytotoxicity on HEK293 cells. These together indicated that the KY44 aptamer could be a potential inhibitor of both WT and KY-mutant HIV-RT.

## 1. Introduction

The reverse transcriptase (RT) enzyme of HIV-1 is the main target for antiretroviral therapy. The HIV-1 RT enzyme is translated from the pol gene and consists of two subunits, p66 and p51, in an asymmetric heterodimer and its multifunction activities include RNA- and DNA-dependent DNA polymerase and RNase H activities. The small subunit, p51, is derived from p66 by proteolytic cleavage of the RNase H domain at the C-terminal and it showed the functions of both an architectural role in the heterodimer and contributes to template/primer binding. The fold of the p66 subunit has been in both the polymerase and RNase H domains [1,2]. The US Food and Drug Administration (FDA) has approved 24 drugs for treating AIDS. Ten drugs target HIV-1 RT in terms of Non-Nucleoside Reverse Transcriptase Inhibitors (NNRTIs) and Nucleoside Reverse Transcriptase Inhibitors (NRTIs), which act with different mechanisms. Conversely, the main reason for treatment failure is the occurrence of resistance-associated mutations in the pol gene of HIV. Many researchers have reported primary drug treatment-induced resistance mutations against NNRTI. A high prevalence of NNRTIs drug mutations was found in K103N, V106M, Y181C, and G190A [3,4]. Ngo-Giang-Huong et al. informed that the mutation of K103N was detected in patients on efavirenz (EFV) more than nevirapine (NVP) treatment. While Y181C was detected with NVP more than EFV [5]. Wang et al. reported NVP induced K103N, Y181C, and G190A mutations [6]. K103N/ Y181C double mutations were associated with resistance to EFV and NVP NNRTI inhibitors [7]. These mutations could lead to enhanced HIV progression and transmission of drug-resistant strains.

HIV-1 research has identified a number of molecular targets for the development of diagnostics and therapy by using aptamers. The short oligonucleotides were applied to drug discovery in terms of oligonucleotide aptamers, single stranded, short RNA or DNA molecules of 20–80 nucleotide in length. Aptamers can form in tertiary structures and bind with high specificity to their targets. Aptamer research has used SELEX technology to identify promising candidates by increased rounds of selection [8,9]. Aptamers can be used to target small/large molecules, complex molecules, agonists, antagonists and be utilized as activators or inhibitors of cell signaling in specific diseases [10,11,12]. Aptamers can be used in various fields such as diagnostics, delivery, bio-imaging, bio-sensing and therapeutics [13,14]. Furthermore, aptamers have been modified for increased binding affinity and more function by reporter molecules, linkers or through immobilization [15]. The FDA approved oligonucleotide therapies in 2017 with six aptamers [16] and focused on four areas: (i) drug-drug interaction (DDI) potential, (ii) organ impairment, (iii) immunogenicity, and (iv) cardiac safety [17]. Recent HIV-1 research has identified several aptamers for the development of therapeutics and diagnostics.

In a previous study, an in silico-designed 16 nt-long RNA aptamer inhibited HIV-1 close to 85% by efficiently interfering with HIV-1 replication [18]. In the anti-HIV-1 integrase aptamer T30175 the individual thymidine forming loops were replaced by 5-hydroxymethyl-2-deoxyuridine residues (H) [19]. Additionally, our research found that aptamers inhibited WT HIV-1 RT activity at an IC50 value of 84.81 ± 8.54 nM and a KD value of 75.10 ± 0.29 nM [20]. This research aimed to isolate potential specific anti-HIV-1 RT DNA aptamers against K103N/Y181C double mutant (KY) HIV-1 RT. The structure of selected DNA aptamer complexed with HIV-1 RTs was characterized.

## 2. Results

### 2.1. Post Selection

To screen the high binding DNA aptamers on HIV-1 RTs (WT and KY), the GNPs-based colorimetric assay was used. This method was used to characterize the DNA aptamers that can complex with HIV-1 RTs with high affinities by detecting the color changes. Generally, GNPs have an optical absorption spectrum around 520 nm and show a red color. Aggregation of GNPs shifts their optical absorption spectrum to a higher wavelength at around 630 nm and making them a purple color [20,21]. The complete reaction contained KY DNA aptamer, HIV-1 RT, GNPs, and NaCl and was performed to compare the binding affinity ratio between WT and mutant. Sixty-one KY DNA aptamers were tested against WT and KY HIV-1 RTs. DNA aptamers were heated and chilled on ice to form the secondary structure before being used. The reactions were measured by a spectrophotometer in the scanning mode and the spectra were collected (Figure 1A). The fold change of 630/520 nm ratios continued to be calculated and Figure 1B shows the ratio of binding affinity of KY DNA aptamers to KY HIV-1 RT. Twenty-eight aptamers changed the color of GNPs to purple with at least a 2.5-fold increase in absorption. This approach is suitable for use as colorimetric systems for the selection of DNA aptamers binding to HIV-1 RTs. Thirteen DNA aptamers (KY1, KY22, KY23, KY24, KY28, KY41, KY42, KY43, KY44, KY56, KY57, KY58, and KY62) that bound to HIV-1 RTs with a high affinity (ranging from 2–4 fold change) were sampled to test for anti-HIV-1 RT activity.

### 2.2. Study of the Function of Aptamers on HIV-1 RTs Activity

The selected KY DNA aptamers were used to determine their activity against HIV-1 RTs. The HIV-1 RTs inhibitory experiment used the EnzChek^®^ Reverse Transcriptase Kit [22]. This experiment was used to determine the activity of DNA aptamers on HIV-1 RTs polymerase function and used 0.5 μM Efavirenz (EFV) as the positive control. Thirteen selected KY DNA aptamers (KY1, KY22, KY23, KY24, KY28, KY41, KY42, KY43, KY44, KY56, KY57, KY58, and KY62) were amplified and heated before being used. Figure 2A represents the relative inhibition of KY DNA aptamers on KY-mutant HIV-1 RT activity. The results show that KY1, KY22, KY23, KY28, and KY44 had more activity than other aptamers by inhibiting the KY-mutant HIV-1 RT activity at 60–85% relative inhibition and more than EFV which showed around 52% relative inhibition. Five KY aptamers were chosen to synthesize. To prove the function of synthesized KY aptamers, five selected KY aptamers were prepared to test against both WT and KY HIV-1 RTs. EFV at a concentration of 0.5 μM was used as the positive control for HIV-1 RTs activity. The results demonstrate that all five synthesized aptamers could inhibit both WT and KY HIV-1 RTs function by providing a % relative inhibition of more than 80% whereas EFV showed 85% and 50% relative inhibition on WT and KY HIV-1 RTs activity, respectively. This experiment confirmed that the 5 selected KY DNA aptamers could inhibit the HIV-1 RTs DNA polymerization function of both WT and KY double mutants. These five synthesized KY aptamers were further used to determine the IC50 values on the DNA polymerase function of HIV-1 RTs.

In a further study, the synthesized KY DNA aptamers were used to determine the IC50 values on HIV-1 RTs DNA polymerase function. The aptamer samples were diluted in the range of 0.46–9000 nM in 3-fold dilution with reaction buffer. The samples were heated and chilled on ice to form the secondary structure. The RT reaction contained 5 mL of each DNA aptamer, 5 mL of 25 nM HIV-1 RTs (WT or KY), and 15 mL of template/primer. Figure 3 shows the IC50 values of the five synthesized DNA aptamers on WT and KY HIV-1 RTs. EFV was used as the positive control. The synthesized KY DNA aptamers showed IC50 in the nanomolar range both of WT and KY HIV-1 RTs (Table 1). KY44 aptamer showed the highest activity on WT and KY HIV-1 RTs DNA polymerase function with IC50 values equal to 45.14 ± 5.75 and 63.82 ± 8.72 nM, respectively. Interestingly, EFV showed IC50 values at 16.03 + 3.83 nM on WT HIV-1 RT and 399.30 + 4.86 nM on KY HIV-1 RT. Five synthesized aptamers (KY1, KY22, KY23, KY28, and KY44) were studied for the binding kinetics by SPR experiment comparing WT and KY HIV-1 RTs.

### 2.3. Affinity Analysis of KY DNA Aptamers and HIV-1 RTs

To characterize the binding kinetics of the HIV-1 RTs-aptamer complexes, an SPR experiment was performed. Generally, Surface Plasmon Resonance or SPR was used to study the interaction between targets immobilized at a solid-liquid interface and partners in solution of the label-free system; and the analysis was performed in real time [23]. The five selected KY aptamers were prepared in SPR running buffer in a 2-fold serial dilution ranging from 125–1000 nM. The results show that all DNA aptamers interacted with both WT and KY HIV-1 RTs. These bindings revealed KD values between 0.55 and 12 μM as shown in Table 2. KY23 displayed the highest binding affinity to both WT and KY HIV-1 RTs. These bindings showed KD values at 5.52 ± 0.4 × 10^−8^ and 1.46 ± 0.005 × 10^−7^, respectively. Moreover, ITC experiments were performed to study the protein-ligand interaction by measuring the heat change of the reaction.

The binding of protein and ligand normally induces a heat change, the interaction particularly provided the reduction in heat; and it is identified as an endothermic property, whereas the increase in heat is termed an exothermic property. Based on the activity of DNA aptamers, KY44 was chosen to study the interaction between aptamer-HIV-1 RTs binding complexes by an NMR experiment.

### 2.4. Investigation of HIV-1 RTs Binding to DNA Aptamer by NMR

Nuclear magnetic resonance or NMR was used to study the interaction between protein and ligand. In a previous study, Thammaporn et al. [24] showed the HIV-1 RT-NNRTIs interactions by NMR. They reported that four methionines (M16, M184, M230, and M357) of HIV-1 RT represented the NNRTI drug binding pocket caused by the chemical shift upon binding to NNRTIs. Our previous report investigated the binding complex of HIV-1 RTs and a DNA aptamer by ^1^H-^13^C heteronuclear single-quantum coherence (HSQC) and showed that the aptamer affected the NNRTI drug binding pocket of HIV-1 RT [20]. In this research, we proposed to study the interaction between KY DNA aptamers isolated from KY double mutant HIV-1 RT by NMR. Figure 4 shows the HSQC spectra of HIV-1 RTs-KY DNA aptamer complexes. WT HIV-1 RT-KY44 aptamer complexes (green spectra) caused a chemical shift in the methionine at the position 357 and had a small effect on M184 (Figure 4A). The complex of KY HIV-1 RT-aptamer displayed a small effect on M184, M230, and M357 as shown in Figure 4B. These positions are related to the NNRTI drug binding pocket, meaning that the KY44 aptamer affected the NNRTI drug binding pocket of HIV-1 RTs.

### 2.5. Toxicity Testing on HEK293 Cells

The KY44 DNA aptamer and EFV were prepared to analyze the cytotoxicity on HEK293 cells. The inhibitors were diluted to 0.076-500 μM in a 3-fold serial dilution. HEK293 cells were treated with HIV inhibitors in a CO_2_ incubator. After 72 h, the culture medium was removed then replaced with Presto Blue (10-fold dilution in culture medium) at 37 °C for at least 1 h. The treated cells were measured at the wavelengths of 570/610 nm and calculated for the percentage of cell survival. The results show that EFV affected HEK293 cells with IC50 values equal to 39.95 ± 2.25 μM; whereas KY44 had an effect at the high concentration with nearly 20% cell death as shown in Figure 5. Therefore, the viral infection was studied by using 20 μM of inhibitors.

### 2.6. Inhibition of HIV Infection

To elucidate that the KY44 DNA aptamer could inhibit HIV infection, an HIV pseudotyped virus system was investigated. Uninfected and pseudo-HIV-1 particle-infected cells were used as the infection control. The viral infections were determined by qPCR [25]. The results revealed that KY44 aptamer also inhibited the infection of the pseudo-HIV-1 particles in HEK293 cells compared with NVP and EFV in a co-culture system (Figure 6). Data were analyzed by Dunnett’s multiple comparisons test (Graphpad Prism 6) and compared between EFV and KY44 (*p* < 0.05). The analyzed data of KY44 could decrease pseudo-HIV-1 particles infecting HEK293 cells when compared to EFV.

## 3. Discussion

HIV-1 RT is the main target for AIDS therapy. The point mutation was increased after treatment with NNRTI drugs. These mutations provided the resistance by interrupting the enzyme-inhibitor interactions by induced conformational change of RT [26]. For instance, K103N is the most common mutation which prevents NNRTIs entry into the NNRTI binding pocket. Another example is Y181C mutation; this residue is important for hydrophobic interaction between HIV-1 RT and NNRTIs inhibitors [27,28]. This study was proposed to identify the new type K103N/Y181C HIV-1 RT inhibitors in terms of DNA aptamers. A total of sixty-one DNA aptamers isolated from K103N/Y181C double mutant HIV-1 RT were studied. These aptamers were screened in terms of the high binding affinity to both WT and KY HIV-1 RTs by a GNPs-based colorimetric assay. This method could be separated into thirteen high binding aptamers to both WT and KY HIV-1 RTs from other aptamers. The aptamers of interest were used to determine their effect on HIV-1 RTs activity. Previously, EnzChek^®^ Reverse Transcriptase Kit-based fluorescent technique was used to measure the anti- HIV-1 RTs inhibitors [22]. The function of thirteen high binding DNA aptamers were identified by EnzChek^®^ Reverse Transcriptase. This method determined the functional DNA aptamers on HIV-1 RT activity. Five aptamers, KY1, KY22, KY23, KY28, and KY44, demonstrated high inhibition of KY-mutant HIV-1 RT DNA polymerase activity at 60–85% relative inhibition, which was higher than EFV (52%) at the same concentration. The five selected DNA aptamers were then synthesized and their function on both WT and KY HIV-1 RTs was again determined. The results confirm that the five synthesized DNA aptamers inhibited HIV-1 RTs DNA polymerase activity both on WT and KY. KY44 aptamer, especially, showed the highest activity. Subsequently the synthesized DNA aptamers were used to measure the IC50 values against WT and KY HIV-1 RTs. The synthesized DNA aptamers showed the IC50 values at the nanomolar level ranging from 45 to 100 nM against WT HIV-1 RT and 60 to 115 nM against KY HIV-1 RT, whereas EFV was used as the positive control. The experiments show IC50 values at 16.03 ± 3.83 nM and 399.30 ± 4.86 nM against WT and KY HIV-1 RTs, respectively. The results show that the selected DNA aptamers, especially KY44, inhibited the HIV-1 RTs DNA polymerase function of WT and KY double mutant. These aptamers did not interact with HIV-1 RTs at the K103N/Y181C position because they presented the same IC50 values on both WT and KY HIV-1 RTs. Previous studies reported that the aptamers specific to HIV-1 RT interacted with HIV-1 RT protein at the substrate-binding site and RNase H domain [29,30,31]. These are the reasons why the KY44 aptamer could inhibit both WT and KY HIV-1 RTs. Therefore, five synthesized DNA aptamers were used to analyze the binding affinity by SPR. SPR is normally used to determine the kinetics binding affinity between ligand and their analyte in real time. This experiment provided the kon, koff, and KD values [32]. This study used HIV-1 RTs as the ligand which was immobilized on an NTA sensor chip and used the synthesized DNA aptamers as analytes. The analyte was injected into the flow cell by increasing the concentration. The results show our synthesized DNA aptamers bound to both WT and KY HIV-1 RTs in the range of 0.06–2 μM and 0.15–2 μM, respectively. Based on the activity of DNA aptamers, KY44 was chosen to study the structure-activity relationship of HIV-1 RTs-DNA aptamer complexes. Evaluation of the binding affinity of the HIV-1 RTs-KY44 aptamer complex was done by ITC experiments. This experiment could not detect the enthalpy change of the complex. In addition, the NMR experiments showed the effect of KY44 aptamer when complexed with WT HIV-1 RT through inducing a chemical shift at methionine residues of M184 and M357. For KY HIV-1, the RT-KY44 aptamer complex showed some small effects on methionine residues at the positions of M184, M230, and M357. These residues represented the NNRTIs drug binding pocket as shown in the previous study [24]. Taken together, the KY44 aptamer might interact with the HIV-1 RTs at the NNRTI drug binding pocket through hydrophobic interaction through which it could induce a conformational change of RT, and inhibited the DNA polymerase function [33]. The selected KY44 aptamer was used to study its cytotoxicity on HEK293 cells and select the inhibitors concentration for anti-HIV infection in comparison with EFV. The results imply that KY44 aptamers were toxic to HEK293 cells at the high concentration (500 μM) with nearly 20% of induced cell death, whereas EFV showed an IC50 value at 39.95 ± 2.25 μM. For anti-HIV infection, pseudo-HIV particles were prepared in a culture medium in the presence or absence of 20 μM EFV or the KY44 aptamer. The results demonstrate that KY44 could inhibit pseudo-HIV particles infection in HEK293 cells by nearly 80%. A further experiment with the DNA aptamer should investigate the mechanism of inhibition on HIV-RT function and pseudo-HIV particle infection.

## 4. Materials and Methods

The isolated DNA aptamers against KY HIV-1 RT were screened by used gold nanoparticles-based colorimetric assay to characterize highly promising KY DNA aptamers. Subsequently, the KY DNA aptamers were used to investigate their function on HIV-1 RTs activity and to characterize the binding affinity. Finally, an NMR experiment was used to determine the structures of HIV-1 RTs–KY DNA aptamer complexes.

### 4.1. DNA Aptamer Preparation

The selected KY DNA aptamers were amplified by specific primers: forward primer 5′-CTTCTGCCCGCCTCCTTCC-3′ and reverse primer 5′AGTGTCCGCCTATCTCGTCTC C-3′ (synthesized by Integrated DNA Technology; IDT Singapore). For the amplification step, 20 μL of the PCR mixture was prepared consisting of 2 μL of 10X reaction buffer, 2 μL of 2.5 mM dNTP mix, 1 μL of 5 μM forward primer, 1 μL of 5 μM reverse primer, 1  μL of each KY DNA aptamer plasmid, and 0.2 μL of DreamTaq™ DNA Polymerase (Thermo Scientific™, Waltham, MA, USA), and the volume was adjusted to 20 μL with nuclease free water. PCR products were purified and dissolved in nuclease free water. The samples were stored at −20 °C prior to use. The synthetic ssDNA aptamers were synthesized by Macrogen (Seoul, Korea) and dissolved in nuclease free water to a stock concentration of 200 μM and stored at −20 °C prior to use. The sequences of the synthetic ssDNA aptamers were as follows:

KY1: AGTAAGACTATAATTCAGTCCTCATTGATTTCCCACATCG

KY22: CACACTAAGGGAGTAGGAATCCGCGGTCACAATGTTCCTG

KY23: CTATAGGGGGTTGCATCTTAGACATTAATGTTCATTTACA

KY28: CCAGAAAGCTCGGGTTCGTTCCACAAATAACACTTGCCC

KY44: GTGCAAGCGTACGAAGGGATAATTCACTCTAGGAGCTTCT

### 4.2. In Vitro Selection of High-Affinity DNA Aptamers for KY-Mutant HIV-1 RT

To select high-affinity KY-mutant HIV-1 RT DNA aptamers, a gold nanoparticles (GNPs)-based colorimetric assay was used. The GNPs-based colorimetric assay followed the procedures of our previous work [20,34]. The complete GNPs (Sigma-Aldrich, Saint Louis, MO, USA) reaction was composed of 5 μL of 100 ng/μL PCR products (DNA aptamers), 37.5 μL of 15 nm GNPs, 5 μL of 50 ng/μL protein target (WT or KY HIV-1 RTs) and 2.5 μL of 2 M NaCl. Each DNA aptamer was heated at 95 °C for 5 min and chilled on ice for 5 min followed by the addition of GNPs, protein target, and NaCl. The samples were incubated at room temperature for 5 min before being transferred to a 1 cm path length quartz cuvette for measurement of the absorbance spectra. The color changes were also detected by spectrophotometer (Varian Cary^®^ 50 UV-Vis Spectrophotometer, Victoria, Australia) in the scanning mode. The absorption spectra changes of GNPs solutions in the presence or absence of targets were determined. The absorbance ratios were calculated from the absorbance of 630 nm divided by 520 nm and compared with or without target protein. A higher ratio of absorbance at 520 nm was obtained from GNPs and at 630 nm was obtained from DNA aptamers complexed with HIV-1 RTs. Fold change of the absorbance was calculated by using the following Equation (1).
(1)$$\mathrm{Fold}\;\mathrm{change}=\frac{\left(A630C/A520C\right)}{\left(A630T/A520T\right)}$$

C = Control: complete reaction without HIV-1 RT

T = Test: complete reaction contained HIV-1 RT

### 4.3. HIV-1 Reverse Transcriptase Inhibition Assay

The inhibition assay was conducted by using the EnzChek^®^ Reverse Transcriptase Assay Kit (Invitrogen™ E-22064, Life Technologies, Carlsbad, CA, USA) following Silprasit et al. [22]. In brief, to conduct the HIV-1 RT inhibitory assay, 5 µL of 2.5 µg/µL of each aptamer sample was pipetted into a 96-well black plate. The negative control consisted of 5 µL of sample buffer, and 5 µL of 2.5 µM Efavirenz as the positive control. After that, 15 µL of template/primer polymerization buffer was added into each well. In the background control the complete HIV-1 RT reaction containing each aptamer and 2 µL of 0.2 M EDTA (stop solution) were used. The reaction was started by adding 5 µL of 25 nM purified recombinant WT or KY HIV-1 RTs and incubated at 25 °C for 30 min. The reactions were stopped by adding 2 µL of 0.2 M EDTA and measured by using the PicoGreen fluorometric method with Tecan Infinite^®^ 200 PRO (DKSH (Thailand) Limited) with an excitation/emission wavelength at 485/535 nm. The inhibitory effects on HIV-1 RTs’ activity were compared by the percentage of relative inhibition, which was measured by using the following Equation (2).
(2)$$\%\;\mathrm{Relation}\;\mathrm{Inhibition}=\frac{\left[\left(\mathrm{RTcontrol}-\mathrm{RTbackground}\right)-\left(\mathrm{RTsample}-\mathrm{RTbackground}\right)\right]\times 100}{\left(\mathrm{RTcontrol}\;-\;\mathrm{RTbackground}\right)}$$

### 4.4. Fifty Percent Inhibitory Concentration (IC50) Determination

For IC50 calculations, each DNA aptamer was prepared in a 3-fold serial dilution with increases through concentrations of 0.46–9000 nM. The reactions were performed in 96-well black plates by adding 5 µL of each KY DNA aptamer, 5 µL of 25 nM purified WT or KY HIV-1 RTs followed by 15 µL of the template/primer polymerization buffer into each well. The reactions were incubated at 25 °C for 30 min. Afterwards, 2 µL of 0.2 M EDTA, the stop solution, was added to stop the reaction. The background control consisted of the complete HIV-1 RT reaction components including each aptamer and 2 µL of 0.2 M EDTA added to stop the reaction. The reaction results were determined by using the PicoGreen fluorometric method. Nonlinear regression dose-response curves plotted percent inhibition and log inhibitor concentration. The 50% inhibition concentrations (IC50) were analyzed using the GraphPad Prism program (GraphPad Software Inc. version 6 access on 1 October 2021, San Diego, CA, USA).

### 4.5. Surface Plasmon Resonance (SPR)

SPR (OpenSPR™; Nicoya Lifesciences, ON, Canada) was used to determine the affinities of the aptamer complexes WT and KY HIV-1 RTs. Measurement of KD values by using OpenSPR™ was performed according to Ratanabunyong et al. [20]. Each DNA aptamer which was used as the analyte was diluted in a 2-fold serial dilution ranging from 125 to 1000 nM in SPR running buffer containing 10 mM HEPES pH 7.5, 150 mM NaCl, 80 mM KCl, and 5 mM MgCl_2_. The WT or KY HIV-1 RTs (ligand) were prepared in SPR running buffer to make a concentration of 1 μM and was used for immobilization on NTA sensor chips (Nicoya Lifesciences; Kitchener, ON, Canada). NTA sensor chips were immobilized with 200 µL of 1 mM WT or KY HIV-1 RTs using a flow rate of 20 µL/min. The increasing concentrations of each DNA aptamer were injected over an NTA sensor chip. The binding was measured by using the association period of 90 s, followed by 180 s of running buffer for dissociation time. All of the SPR experiments were completed at 25 °C in running buffer using a flow rate of 50 µL/min. Binding parameters were calculated by using TraceDrawer Software version 1.6.1 access on 12 July 2021 (Ridgeview Instruments AB, Uppsala, Sweden) with the model of 1:1 binding, based on the theory that the immobilized target protein binds to the aptamers in a 1:1 ratio.

### 4.6. Nuclear Magnetic Resonance (NMR)

To determine the structure-activity relationship of HIV-1 RT complexed with DNA aptamers, an NMR experiment was performed following Thammaporn et al. [24]. NMR is an effective method for analyzing HIV-1 RT binding to drugs. The results show the ^1^H-^13^C heteronuclear single-quantum coherence (HSQC) data of HIV-1 RT-NNRTIs. The ^1^H-^13^C HSQC spectra of WT or KY HIV-1 RTs labeled at methyl-^13^C-methionine on the p66 subunit were measured in the presence and absence of DNA aptamers. The HIV-1 RT and DNA aptamers were prepared in 10 mM Tris-d11 buffer (pD 7.6) containing 200 mM KCl, 1.5 mM sodium azide, and 4 mM MgCl_2_. Twenty-eight micromolar HIV-1 RT was complexed with 140 mM DNA aptamers (1:5 molar ratio). The reaction was measured by using an AVANCE800 (Bruker BioSpin; Karlsruhe, Germany) spectrometer supplied with a cryogenic probe. The spectra data were prepared and analyzed by the Topspin 3.2 (Bruker BioSpin; Karlsruhe, Germany) and SPARKY 3.115 programs (San Francisco, CA, USA).

### 4.7. Cytotoxicity on HEK293 Cells

The cytotoxicity assay was performed by following Engelberg et al. [35] (Engelberg, Netzer, Assaraf, and Livney, 2019). HEK293 cells (ATCC, CRL-1573) were cultured in DMEM containing 10% FBS and 1% antibiotics. Cells were seeded in a 96-well plate at the density of 1 × 10^5^ cells/mL and incubated at 37 °C in an atmosphere containing 5% CO_2_ overnight. The KY44 aptamer and EFV were diluted in DMEM complete medium to make the concentration in the range of 0.076–500 μM (3-fold serial dilution). The culture medium was removed and fresh medium containing KY44 aptamer or EFV at the various concentrations were added into the desired wells. The treated cells were incubated at 37 °C in an atmosphere containing 5% CO_2_ for 72 h. After that, culture medium was removed and replaced with Presto Blue^®^ Cell Viability Reagent (Carlsbad, CA, USA) diluted to 1:10 with the complete medium. Cells were incubated in a CO_2_ incubator for at least 30 min and the absorbance of the reagent at 570 and 610 nm was measured. The percent cell survival was calculated by using the following Equation (3).
(3)$$\%\;\mathrm{Cell}\;\mathrm{survival}\;=\frac{\left(A570-610\;\mathrm{test}\right))}{\left(A570-610\;\mathrm{untreated}\right)}$$

### 4.8. Pseudo-HIV-1 Particle Infection

The lentiviral system was used to propagate the pseudo-HIV-1 particles by co-transfected pfNL43-dE-EGFP (which was a gift from Mario Chin (Addgene plasmid # 36865) and was a gift from Didier Trono (Addgene plasmid # 12259) into HEK293 cells by using the lentiviral system following Tiscornia et al. [36]. HEK293 cells were seeded in a 100 mm dish at the density of 5 × 10^6^ cells and incubated at 37 °C in an atmosphere containing 5% CO_2_ overnight. Cells were transfected with 10 μg pfNL43 and 5 µg pMD2.G with lipofectamine 3000 and incubated at 37 °C in an atmosphere containing 5% CO_2_ for 72 h. The pseudo-HIV-1 particles were harvested and stored at −80 °C before use. For the anti-HIV infection determination, HEK293 cells were seeded in 6-well plates at the density of 5 × 10^5^ cells/well and incubated overnight at 37 °C in an atmosphere containing 5% CO_2_ overnight. The culture medium was removed and cells were infected with 400 µL of pseudo-HIV-1 particles containing 20 µM of the KY44 aptamer, NVP, or EFV into desired wells. The infected cells were incubated for 1 h by rocking every 15 min. The infected cells were amended with DMEM complete medium and incubated at 37 °C in an atmosphere containing 5% CO_2_ for 72 h. The infected cells were harvested and the viral DNA was extracted [37]. The percentage of pseudo-HIV-1 particle infection was determined by used qPCR with the HIV LTR specific primer (forward primer 5′-GCGCTTCAGCAAGCCGAGTCCT-3′ and reverse primer 5′-CACACCTCAGGTACCTTTAA GA-3′) [25].

## 5. Conclusions

This study determined the functional DNA aptamers on HIV-1 RT activity. Five aptamers, KY1, KY22, KY23, KY28, and KY44, demonstrated high inhibition of KY-mutant HIV-1 RT DNA polymerase activity at 60–85% relative inhibition. The IC50 values at the nanomolar level ranging from 45 to 100 nM against WT HIV-1 RT and 60 to 115 nM against KY HIV-1 RT. The SPR results show our five DNA aptamers bound to both WT and KY HIV-1 RTs in the range of 0.06–2 µM and 0.15–2 µM, respectively. The KY44 was chosen to study the structure-activity relationship of HIV-1 RTs-DNA aptamer complexes. The NMR experiments showed the effect of KY44 aptamer when complexed with WT HIV-1 RT through inducing a chemical shift at methionine residues of M184 and M357. For KY HIV-1, the RT-KY44 aptamer complex showed some small effects on methionine residues at the positions of M184, M230, and M357. Moreover, KY44 could inhibit pseudo-HIV particle infection in HEK293 cells with nearly 80% inhibition and showed low cytotoxicity on HEK293 cells. These together indicated that the KY44 aptamer could be a potential inhibitor of both WT and KY-mutant HIV-RT. A further experiment should investigate the mechanism of DNA aptamer inhibition on HIV-RT and pseudo-HIV particle infection.

## Figures and Tables

**Figure 1 molecules-27-00285-f001:**
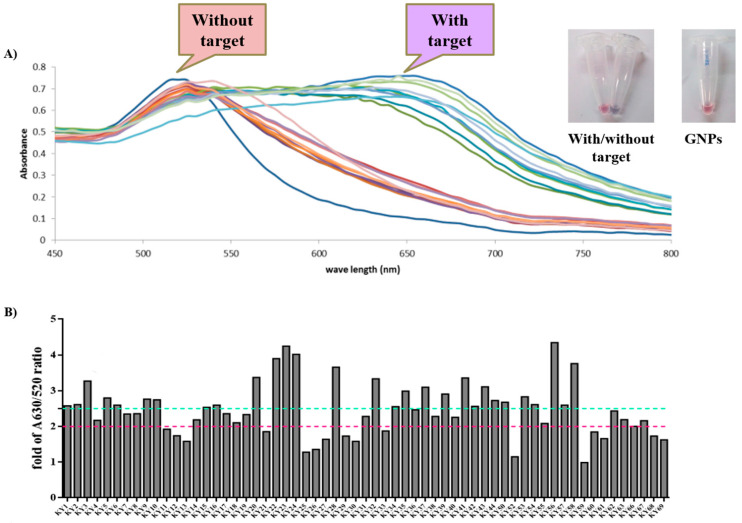
GNPs-based colorimetric results. (**A**) The spectra of HIV-1 RTs in the presence and absence of KY DNA aptamers. Each color line represent for GNPs-based colorimetric assay in with or without HIV-1 RT. (**B**) Fold change of DNA aptamers when complexed with KY HIV-1 RT. Green line at cutoff 2.5-fold change and pink line at cutoff 2.0-fold change.

**Figure 2 molecules-27-00285-f002:**
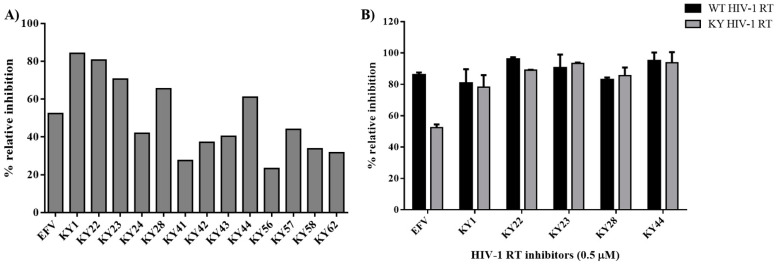
HIV-1 RTs DNA polymerase inhibition experiments. (**A**) Percent relative inhibition of amplified 13 KY DNA aptamers on KY-mutant HIV-1 RT function. (**B**) Percent relative inhibition of five synthesized KY DNA aptamers on both WT and KY HIV-1 RTs function. Data are represented as the mean ± SD (*n* = 3).

**Figure 3 molecules-27-00285-f003:**
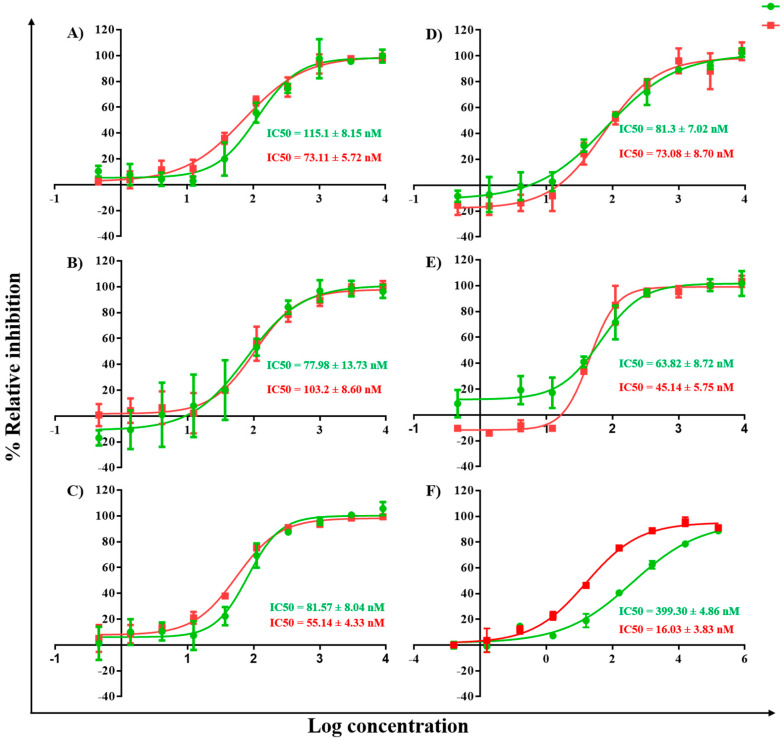
IC50 values of KY DNA aptamers on HIV-1 RTs polymerase activity. The red and green lines represent WT and KY HIV-1 RTs activity, respectively, in the presence of DNA aptamers. (**A**) KY1 (**B**) KY22 (**C**) KY23 (**D**) KY28 (**E**) KY44 and (**F**) EFV IC50.

**Figure 4 molecules-27-00285-f004:**
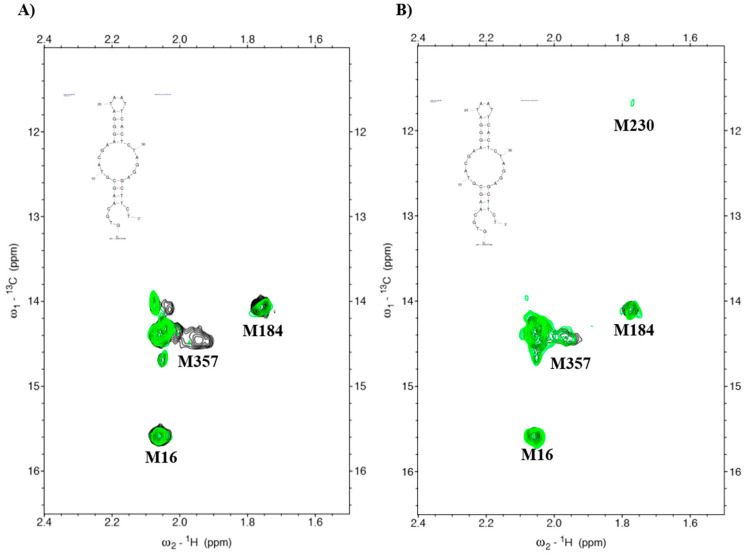
HSQC spectra of HIV-1RTs-aptamers. Black represents the HIV-1 RTs apo form. Green spectra represents the HIV-1 RTs-aptamer complexd. (**A**) WT HIV-1 RT complexed to the KY44 aptamer. (**B**) KY-mutant HIV-1 RT complexed to the KY44 aptamer.

**Figure 5 molecules-27-00285-f005:**
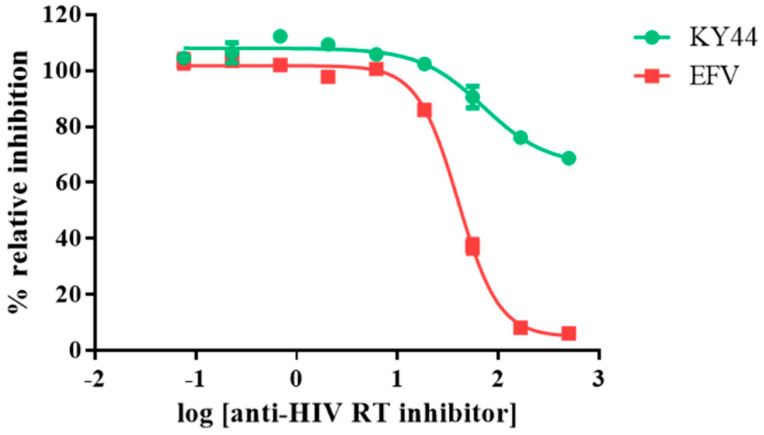
Cytotoxicity testing of HIV inhibitors on HEK293 cells survival. The green line represents KY44 and the red line represents EFV. *Y* axis showed the % relative inhibition to untreated cells. *X* axis showed log concentration of HIV-1 RTs inhibitors. At the high concentration, EFV and KY44 were toxic to HEK293 cells at around 100 and 20% relative inhibition, respectively. Data are represented as the mean ± SD (*n* = 3).

**Figure 6 molecules-27-00285-f006:**
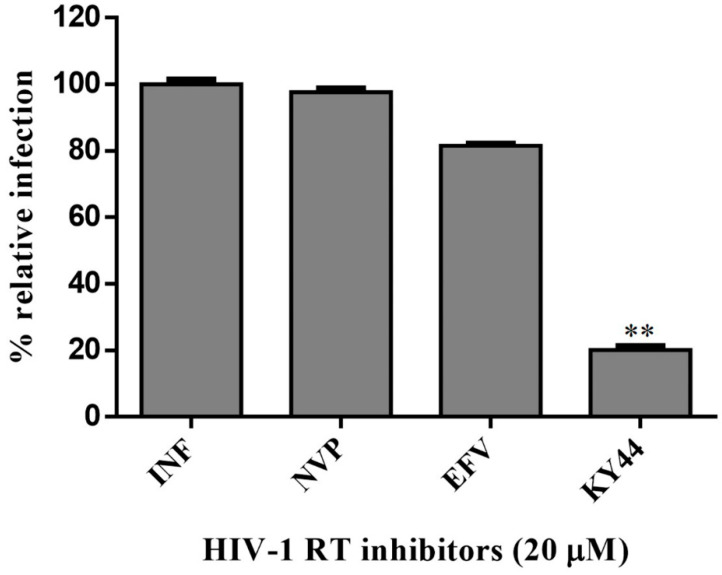
Percentage of the relative infection of pseudo-HIV-1 particles in HEK293 cells. Data are represented as the mean ± SD (*n* = 3). ** is the significance of KY44 decrease when compared to EFV.

**Table 1 molecules-27-00285-t001:** IC50 values of synthesized KY DNA aptamers on HIV-1 RTs DNA polymerase function.

Sample No.	IC_50_ (nM)	
	WT HIV-1 RT	KY HIV-1 RT
KY1	73.11 ± 5.72	115.10 ± 8.15
KY22	103.20 ± 8.60	77.98 ± 13.73
KY23	55.14 ± 4.33	81.57 ± 8.04
KY28	73.08 ± 8.70	81.30 ± 7.02
KY44	45.14 ± 5.75	63.82 ± 8.72
EFV	16.03 ± 3.83	399.30 ± 4.86

Data are represented as the mean ± SD (*n* = 3).

**Table 2 molecules-27-00285-t002:** Binding affinity of HIV-1 RTs-aptamer complexes.

Sample No.	WT HIV-1 RT			KY HIV-1 RT		
	k_on_ (1/(M×s))	k_off_ (1/s)	K_D_ (M)	k_on_ (1/(M×s))	k_off_ (1/s)	K_D_ (M)
KY1	4.89 ± 0.1 × 10^4^	1.06 ± 0.0001 × 10^−1^	2.17 ± 0.04 × 10^−6^	1.38 ± 0.1 × 10^4^	2.99 ± 0.001 × 10^−2^	2.16 ± 0.2 × 10^−6^
KY22	1.01 ± 0.002 × 10^5^	1.04 ± 0.00003 × 10^−1^	1.04 ± 0.002 × 10^−6^	7.27 ± 9.49 × 10^3^	8.93 ± 0.006 × 10^−2^	1.23 ± 2.27 × 10^−5^
KY23	5.22 ± 0.2 × 10^4^	2.88 ± 0.1 × 10^−3^	5.52 ± 0.4 × 10^−8^	6.33 ± 0.02 × 10^4^	9.26 ± 0.003 × 10^−3^	1.46 ± 0.005 × 10^−7^
KY28	6.48 ± 0.05 × 10^4^	6.62 ± 0.0008 × 10^−2^	1.02 ± 0.008 × 10^−6^	1.60 ± 0.03 × 10^4^	2.65 ± 0.0003 × 10^−2^	1.66 ± 0.03 × 10^−6^
KY44	1.77 ± 9.08 × 10^5^	3.08 ± 0.004 × 10^−1^	2.15 ± 0.4 × 10^−6^	3.47 ± 0.01 × 10^4^	5.41 ± 0.0001 × 10^−2^	1.56 ± 0.004 × 10^−6^

Data are represented as the mean ± SD (*n* = 3).

## Data Availability

Data is contained within the article.

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
