# Peer review of "Biophysical Characterization of Novel DNA Aptamers against K103N/Y181C Double Mutant HIV-1 Reverse Transcriptase"

_molecules, 2022, doi:10.3390/molecules27010285_

Round 1
Reviewer 1 Report
In the current study, authors isolate a DNA aptamer that targets reverse transcriptase of HIV-1 (HIV-1 RT), which in in-vitro setting works effectively against both the wild-type and a double mutation K103N/Y181C in HIV-1 RT. It is a systematic study, where the authors start by screening 61 DNA aptamers and narrow down to 5 aptamers that are more efficacious than Efavirenz (EFV), an HIV-1 RT inhibitor that is currently used in the clinics to treat HIV-1. Out of the five aptamers, one aptamer KY-44 was further analyzed with NMR to study its interaction with HIV-1RT, followed up by in vitro toxicity studies in HEK cells. KY44 was found to be less toxic on HEK cells compared to EFV, paving way to further studies. The study is significant considering the ongoing work in aptamers and their relevance in overcoming drug resistance from mutations. While, the study is systematic, the quality of writing and presentation needs significant improvement to improve the quality of the manuscript, which I believe will further improve the impact of the work. Here are some comments; in general some sections of the results emphasize on explaining the experimental detail that the gist of result is lost. Secondly, there is little statistical details in the manuscript.
Result:
Section 2.1 pg. 89-97: The first part of the section reads like methods section. Can authors state just the results and move the rest to method section. It is difficult to comprehend the results with so much of methods described here. Can authors explain the significance of 2.5-fold increase threshold?
Section 2.2: Pg.4 line 145, what do error values indicate (mean +/- S.D., as mentioned in the table)? Did authors compare the values between the two groups (wild-type Vs mutant) to conclude that 5 aptamers inhibit both groups similarly, since there is no report of a statistical test.
Section 2.4: can authors comment on why the affinity (Kd) of aptamers to HIV-1 reverse transcriptase are in the order of uM (table. 2), while the efficacy (IC50) is in nM range (table. 1)?
Section 2.5: Firstly, there is no reference to figure (fig. 5) in this section. It is unclear what are the units of the x-axis of Fig. 5 other than the fact that they are plotted on a log scale, and there is no mention of what is the % inhibition relative to either in the figure or legend.
Section 2.6: Again, most of the first paragraph belongs to the methods section. KY44 seems to provide significant protection against infection of HIV-1 particles. However, there are no error bars, can authors mention if this data was replicated, with statistical significance compared to EFV.
Author Response
Response to Reviewer 1
Comment 1 Section 2.1 pg. 4 line 91-93
The first part of the section reads like methods section. Can authors state just the results and move the rest to method section. It is difficult to comprehend the results with so much of methods descript here. Can authors explain the significance of 2.5-fold increase threshold?
Answers
Thank you for the suggestion. We change the sentence in line 91-93 as below
“This research aimed to isolate potential specific anti-HIV-1 RT DNA aptamers against K103N/Y181C double mutant (KY) HIV-1 RT. The structure of selected DNA aptamer complexed with HIV-1 RTs was characterized.”
And move the sentence “The isolated DNA aptamers against KY HIV-1 RT were screened by used gold nanoparticles based colorimetric assay to characterize highly promising KY DNA aptamers. Subsequently, the KY DNA aptamers were used to investigate their function on HIV-1 RTs activity and to characterize the binding affinity. Finally, an NMR experiment was used to determine the structures of HIV-1 RTs–KY DNA aptamer complexes.” To material and methods in pg. 16 line 308-312
The significance of 2.5-fold increase threshold.
The GNPs-based colorimetric assay was chose to characterize the binding and non-binding DNA aptamers to target proteins (HIV-1 RT). If the DNA aptamers complex to HIV-1 RT the color of GNPs were change to purple and gave the OD to 630nm whereas the control, without target protein, showed in red color and gave OD around 520nm. The ratio of 630/520 of DNA aptamers when complex with the HIV-1 RT were calculated and the 2.5-fold increase in absorption was enough to separate the binding and non-binding DNA aptamers to HIV-1 RT.
Comment 2 Section 2.2 pg. 7 line 145
What do error values indicat (mean +/- S.D., as mentioned in the table)? Did authors compare the values between two grouos (wild-type Vs mutant) to conclude that 5 aptamers inhibit both grops similarly, since there is no report of a statistical test.
Answers
Thank you for the suggestion. The error values indicate mean ± SD (n=3).
The statistical test by compared between wild-type vs mutant showed no significant by used Mann Whitney test (two-tailed P value) as shown in the table.
|
HIV-1 RT inhibitors |
P value |
Significantly different? (P < 0.05) |
|
EFV |
0.3333 |
ns |
|
KY1 |
0.7 |
ns |
|
KY22 |
0.1 |
ns |
|
KY23 |
0.7 |
ns |
|
KY28 |
0.4 |
ns |
|
KY44 |
0.9 |
ns |
Comment 3 Section 2.4
Can authors comment on why the affinity (Kd) of aptamers to HIV-1 reverse transcriptase are in the order of uM (table. 2), while the efficacy (IC50) is in nM range (table. 1)?
Answers
IC50 is used to determine the inhibitory concentration of 50% while Kd measures the dissociated components and the equilibrium between the ligand-protein complex. These values obtained are highly dependent on the mechanism of inhibition and the measurement conditions. Several time from many literature these IC50 and Kd are not equally. However, it can see the relative relationship from these values.
Comment 4 Section 2.5 pg. 12 line 221
Firstly, there is no reference to figure (fig. 5) in this section. It is unclear what are units of the x-axis of Fig. 5 other than the fact that they are plotted on the log scale, and there is no mention of what is the %inhibition relative to either in the figure or legend.
Answers
Thank you for pointing this out. We already “add reference of Figure 5 in line 221.”
and line 222-225
Unit of X axis is the log concentration of HIV-1 RTs inhibitors and Y axis is the % relative inhibition to untreated cells.
“Y axis showed the % relative inhibition to untreated cells. X axis showed log concentration of HIV-1 RTs inhibitors. At the high concentration of EFV and KY44, were toxic to HEK293T cells at around 100 and 20 % relative inhibition, respectively.”
Comment 5 Section 2.6
Again, most of the first paragraph belongs to the methods section. KY44 seems to provide significant protection against infection of HIV-1 particles. However, there are no error bars, can authors mention if this data was replicated, with statistical significance compared to EFV.
Answers
The first paragraph was moved to material and methods on pg. 21 section Pseudo-HIV-1 particle infection.
We add error bar and show statistical significance compared to EFV in Figure 6.
We add the sentence in pg. 13 line 232-235 as below
“Data were analyzed by Dunnett's multiple comparisons test (Graphpad Prism 6) and compared between EFV and KY44 (p < 0.05). The analyzed data of KY44 could decreased pseudo-HIV-1 particles infect to HEK293 cells when compared to EFV.”
Reviewer 2 Report
row 60 ==> I think it should be SELEX technology and not SELEX only
row 66 ==> bio-sensing and not "bio sensing"
row 201 & 255 ==> IC50, 50 should be in subscript
alla nano prefixes seems n and not really the Greek letter "nano". It should be: hM or nM?
row 324, 343 and 403 something weird with the line
Author Response
Response to Reviewer 2
Comment 1
row 60; I think it should be SELEX technology and not SELEX only
Answer
Page 4, row 71
Thank you for pointing this out. The reviewer is correct, and we have change “SELEX to SELEX technology.”
Comment 2
row 66; bio-sensing and not "bio sensing"
Answer
Page 4, row 77
As suggested by the reviewer, we have change “bio sensing to bio-sensing.”
Comment 3
row 201 & 255; IC50, 50 should be in subscript
Answer
Thank you for pointing this out. The IC50, “50 has been corrected on subscript.”
Comment 4
all a nano prefixes seems n and not really the greek letter "nano". It should be: hM or nM?
Answer
All a nano prefixes should be “n”
Comment 5
row 324, 343 and 403 something weird with the line
Answer
Page 17, row 344; Page 18, row 363; Page 21, row 425
Thank you for the reviewer suggestion. We have already adjust the lines into the correct position.